# KVTQ: Compressing the KV Cache to Hardware Efficient Ternary Digits by Fine-Grained Dynamic Quantization

## Abstract

Large language models(LLMs) exhibit capabilities beyond expectations in various NLP tasks. Since the inference of LLM consumes huge resources, optimizing the inference process of LLM is of great significance to promote the application of LLM. In the text generation process, caching the key-value embeddings (KV cache) for subsequent generation process is a basic optimization method. However, huge size of the KV cache limits the inference batch size. Compressing the space occupied by the cached key-value embeddings can enlarge the batch size of LLM inference to improve throughput. Besides, based on the analysis of the usage mode of the KV cache, we find compressing the KV cache to ternary digits can not only compress the space occupied by the KV cache, but also greatly reduce the required multiplication operation in the attention block. Combined with the numerical features of the KV cache, we propose KVTQ, a method which compresses the KV cache to hardware efficient ternary digits. We validate our KVTQ method on different series of LLMs and get the conclusion that the KVTQ method which compresses the KV cache to ultra-low bits can still preserve the model quality.

## 1 Introduction

Pre-trained generative models (Brown et al., 2020; Du et al., 2022; Zeng et al., 2022; Zhang et al., 2022; Chowdhery et al., 2022; Touvron et al., 2023; OpenAI, 2023; Anil et al., 2023), also known as large language models(LLMs), have demonstrated capabilities beyond expectations in various NLP tasks. Due to the huge amount of parameters and calculations of the LLMs, the cost of LLM inference is high. And improving the inference efficiency of LLMs will help people access applications based on LLMs at a lower cost.

LLM inference is an autoregressive process, the next generated token is determined by the input prompt and all previously generated tokens. The key-value embeddings in the attention block are stored to avoid repeated calculation. Using LLaMA-65B (Touvron et al., 2023) as an example, at batch size 128 and sequence length 2048, the KV cache requires 640GB memory. In the text generation process, every time a new token is generated, all weights of the model needs to be loaded. And increasing the batch size can reduce the cost of loading the model and significantly improve the throughput (Pope et al., 2023; Sheng et al., 2023). Unfortunately, the KV cache size is linearly correlated with the batch size used in inference. With the same hardware and the same model, the maximum supported batch size is almost linearly amplified by the compression ratio of the KV cache. So compressing the KV cache is an effective method to improve the efficiency of LLM inference.

Quantization (Frantar et al., 2022; Yao et al., 2022; Lin et al., 2023; Dettmers et al., 2022; Xiao et al., 2023; Yao et al., 2023) has been widely studied in compressing the weights of LLMs. Quantization aware training(QAT) requires inserting quantization operations into the neural network before training or fine-tuning. Because training LLMs is expensive and the training process is complex, it is clear that QAT is not suitable for LLMs. Post training quantization (PTQ) means quantizing the weights or quantizing both the weights and the activations without training involved. For LLMs, directly quantizing both the weights and the activations simultaneously results in unacceptable performance degradation (Dettmers et al., 2022). And weights only quantization is widely used in compressing

the weights of LLMs (Frantar et al., 2022; Lin et al., 2023). However, quantizing the KV cache is rarely studied. For text generation applications, the key-value embeddings are calculated and cached once, but the cache KV is used for every subsequent query. The usage mode of the KV cache determines that we can use a method similar to the weights only quantization to quantizing the KV cache. The difference is that the information required to quantize the weights is contained in the weights themselves whereas the key-value embeddings are calculated dynamically. Since the length of the output sequence is dynamic, it is difficult to choose a proper dimension to split the KV cache and find suitable example inputs to do fine-grained group-wise quantization calibration offline.

When the KV cache is compressed by weights only quantization, dequantization is required every time it is used. For $A \times B$, when either A or B contains only ternary digits, the matrix multiplication can be completed by addition and subtraction operations. If we compress the KV cache to ternary digits, dequantization is no longer needed and the subsequent matrix multiplication can be implemented by addition and subtraction.

Inspired by the above observation, we propose the fine-grained dynamic quantization method to compress the KV cache to ternary digits, called KVTQ[1]. To our knowledge, we are the first to show that the KV cache of LLMs can be compressed to hardware efficient ternary digits with negligible increase in perplexity. First we summarize the related works in section 2. In section 3, we analysis the numerical features of the KV cache and introduce our KVTQ menthod. In section 4, we systematically evaluate the KVTQ method on different series of LLMs and certificate that compressing the KV cache to hardware efficient ternary digits by the KVTQ method result in negligible accuracy degradation. We briefly summarize our work and discuss future work in section 5.

## 2 RELATED WORK

Quantization (Krishnamoorthi, 2018; Wu et al., 2020; Gholami et al., 2022) is widely used to optimize the inference of AI models. Quantization is divided into two categories: Quantization aware training(QAT) and post-training quantization(PTQ). For QAT, users need to modify the training process and simulate the quantization operation in the training process. As mentioned earlier, the cost of training LLMs is expensive so QAT is rarely used in LLMs.

For LLMs, PTQ is widely studied to optimize the inference process. Yao et al. (2022); Dettmers et al. (2022); Xiao et al. (2023) try to quantize both the weights and the activations. Based on the analysis of the numerical features of the LLMs, ZeroQuant (Yao et al., 2022) uses group-wise quantization for weights and token-wise quantization for activations. Their method works for small models but fails to keep accuracy on large models. LLM.int8 (Dettmers et al., 2022) demonstrates that LLMs with model size greater than 6.7B have outliers in the feature maps. Quantizing features maps which contain outliers to INT8 will result in unacceptable performance degradation. They solve this problem by keeping the outliers as FP16 and quantizing the rest features to INT8. However, since the outliers are randomly scattered in the feature maps, it is difficult to design an efficient implementation for LLM.int8. SmoothQuant (Xiao et al., 2023) smooths the feature maps of LLMs by a diagonal matrix and simultaneously the weights are sharped by the inverse matrix of the diagonal matrix. By doing so, the accuracy drop caused by the outliers is alleviated. The methods mentioned above can quantize the feature maps to INT8.

GPUs are the most commonly used device in LLM inference. Using GPU as an example, in the scenario of LLM inference, the bottlenecks are the global memory size of GPU and the GPU bandwidth from global memory to register/shared memory. So weight only PTQ (Frantar et al., 2022; Lin et al., 2023) is widely used to compress the weights of LLMs to reduce the memory compustion and the required bandwidth. GPTQ (Frantar et al., 2022) propose a method which uses the second-order information to compensate the error introduced by quantization. GPTQ can quantize the weights to 3/4bit with acceptable loss of accuracy. AWQ (Lin et al., 2023) find that salient weights are mojored by activation and preserving salient weights in FP16 can improve the performance of LLMs with weigths only quantization. Based on this observation, AWQ propose an activation-aware scaling method to reduce the loss of precision involved by quantization.

The key-value embeddings are intermediate feature maps. The works (Yao et al., 2022; Dettmers et al., 2022; Xiao et al., 2023) which try to quantize the feature maps mentioned above can only

---

[1]This name means compressing the KV cache to Ternary digits by fine-grained dynamic Quantization.

compressing the feature maps to INT8. Quantizing the KV cache to ultra-low bits has rarely been studied. FlexGen (Sheng et al., 2023) is a high-throughput generation engine for LLMs with single GPU. They applies quantization to the KV cache and they mentioned that compressing the KV cache to 3 bits cannot preserve accuracy.

Because of limited expression ability of ultra-low bit digits, compressing the neural network to ultra-low bits is a challange task. But due to the attractive hardware efficient propery, ternary neural networks (Alemdar et al., 2017; Chen et al., 2021) and binary neural networks (Lin et al., 2017; Liu et al., 2018) have been widely studied. LLM inference requires huge amount of memory and computing resources, it is promessing if we could compress the KV cache into binary digits or ternary digits.

Sparsification is another way to compressing the KV cache besides quantization. Scissorhands (Liu et al., 2023) find that the attention maps with high scores are repeated. They compresses the KV cache by only saving the key-value embeddings with high attention scores. Optimizing the model structure can also compress the KV cache. Shazeer (2019) uses the multi-query attention to replace the multi-head attention to reduce the memory consumption of the KV cache.

Up to now, there is no work to systematically analyze the characteristics of the KV cache and compression of the KV cache has rarely been studied. We attempt to fill in the gaps and we hope our work will inspire further research on the KV cache compression.

## 3 APPROACH

In this section we first review the basic concepts of quantization. Then we analyze the fitness between the fine-grained dynamic quantization method and the KV cache compression. Next we analyze the numerical features of the KV cache of different series of LLMs from different perspectives. We use the validation set of the wikitext-2 data set as the inputs to analysis the numerical feature of the KV cache. Inspired by the usage mode of the KV cache, we propose an extreme compression strategy that quantizes the key-value embeddings to ternary digits. Finally based on our analysis and observation, we propose the KVTQ method.

### 3.1 QUANTIZATION BASIC

We first quickly review the quantization operation. The general form of uniform quantization operation can be expressed as:

$$\mathbf{X}^q = \lceil \frac{\mathbf{X}}{\Delta} \rfloor + Z \tag{1}$$

$$\Delta = \frac{max(\mathbf{X}) - min(\mathbf{X})}{2^N - 1}, Z = -2^{N-1} - \lceil \frac{min(\mathbf{X})}{\Delta} \rfloor \tag{2}$$

where $\lceil \rfloor$ indicates rounding to the nearest integer, $\mathbf{X}$ is a floating-point tensor, $\mathbf{X}^q$ is the quantized tensor, $\Delta$ is the quantization step, $Z$ is the zero point, and $N$ is the number of bits. The above expression is also known as asymmetric quantization.

### 3.2 DYNAMIC OR STATIC QUANTIZATION

The KV cache is a 4D tensor with shape $[B, NH, S, HS]$, where $B$ is the batch size, $NH$ is the number of heads, $S$ is the sequence length, $HS$ is the head size. For text generation applications, $S$ is a dynamically changing number and determined by the input prompt and the random sampling operations in the generation process.

Fine-grained group-wise quantization is widely used in quantizing transformer-based models (Shen et al., 2020; Frantar et al., 2022; Lin et al., 2023). This method has been proved to be an efficient way to maintain the accuracy of LLMs. TensorRT-LLM [2] uses static quantization to compress the

---
[2]https://developer.nvidia.com/tensorrt-llm-early-access

KV cache. They use per-tensor quantization step $\Delta$ to quantize the KV cache. And the quantization step $\Delta$ is calculated offline with given example inputs. Due to the dynamic nature of $S$, it is difficult to choose a proper dimension to split the KV cache to do fine-grained group-wise static quantization for LLMs.

Unlike static quantization, which should compute the quantization step $\Delta$ offline, dynamic quantization computes the quantization step $\Delta$ in the forward propagation process. Since the quantization step $\Delta$ is obtained online, we can avoid the difficulty in quantization granularity selection which is caused by the dynamic nature of $S$. We can use per-token, per-head or even sub-head granularity to split the KV cache for fine-grained group wise dynamic quantization. For per-token quantization, we calculate a $\Delta$ with $NH \times HS$ values. And for per-head quantization, we calculate a $\Delta$ with $HS$ values.

For multi-head attention, the KV cache is continuous in the head dimension. When we use the Nvidia GPU as a device for LLM inference, the data of a head can be placed in the share memory of one block, which lead to efficient implementation of the per-head quantization operation. And obviously the finer the granularity, the higher the accuracy. However, when the granularity is very small, the space occupied by storing the quantization step $\Delta$ cannot be ignored. Considering the implementation efficiency and compression ratio, per-head quantization is an appropriate choice.

### 3.3 SYMMETRIC OR ASYMMETRIC QUANTIZATION

Symmetric and asymmetric quantization have been extensively studied (Krishnamoorthi, 2018; Wu et al., 2020; Gholami et al., 2022). Symmetric quantization is a special case of asymmetric quantization. When we set $max(\mathbf{X})$ and $min(\mathbf{X})$ in equation 2 as $max(abs(\mathbf{X}))$ and $-max(abs(\mathbf{X}))$, asymmetric quantization becomes symmetric quantization. Asymmetric quantization obviously has better mapping range than symmetric quantization. However, for both the quantization and dequantization processes, asymmetric quantization requires greater computational cost compared to symmetric quantization. Besides, asymmetric quantization also requires extra space to store zero points. For the KV cache quantization problem, which one is more suitable, symmetric quantization or asymmetric quantization, is a question that needs to be studied.

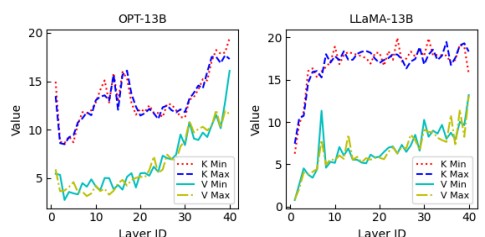

Figure 1: In this figure, we plot the absolute values of the maximum and minimum values of the KV cache for each layer of OPT-13B and LLaMA-13B. For both models, the absolute values of the maximum and minimum values are of the same order of magnitude.

Usually, the key-value embeddings are the result of linear mapping of normalized tensors (Zhang et al., 2022; Touvron et al., 2023). Unlike outputs of activation layers (Nair & Hinton, 2010; Hendrycks & Gimpel, 2016), intuitively, the key-value embeddings are more friendly to symmetric quantization. Inspired by the above analysis, we counted the maximum and minimum values of the KV cache of different models.

As show in figure 1, we find that the absolute values of the maximum and minimum values of the KV cache are in the same order of magnitude. There will not be much difference in the mapping range between symmetric quantization and asymmetric quantization. So we use symmetric quantization for better efficiency and less memory consumption. As a special case of equation 1 2, the symmetric quantization process can be expressed as:

$$\mathbf{X}^q = \lceil \frac{\mathbf{X}}{\Delta} \rfloor, \Delta = \frac{max(|\mathbf{X}|)}{2^{N-1} - 1} \tag{3}$$

Symmetric quantization is also more friendly for us to further quantize the KV cache into hardware efficient ternary digits, which we will discuss in section 3.5.

### 3.4 KV CACHE HAVE DIFFERENT DYNAMIC RANGES

The key and value embeddings are used in different position of the attention block. It is instinctive to consider that their sensitivities to quantization are also different. The key embeddings are used to calculate the attention score. As show in figure 4a, there is a softmax to normalize the product result of each new query and the cached key embeddings. The softmax is smoothed with temperature $T = \sqrt{HS}$. When the head size $HS = 128$, the temperature $T = 11.31$. Intuitively, we think the key embeddings are less sensitive to quantization and can be compressed to fewer bits.

The quantization error depends on the quantization step $\Delta$. And the smaller the quantization step $\Delta$, the smaller the quantization error. As referred in equation 3, the maximum absolute value determines the $\Delta$ used to quantize the KV cache. We counted the maximum absolute value of the KV cache of two series of LLMs. As show in figure 2, the key embeddings have wider ranges. For OPT series models, the average maximum absolute value of the key embeddings is about 2.13 times of the value embeddings. For LLaMA series models, it is 2.57 times. Usually, more bits are required to quantize feature maps with larger value ranges.

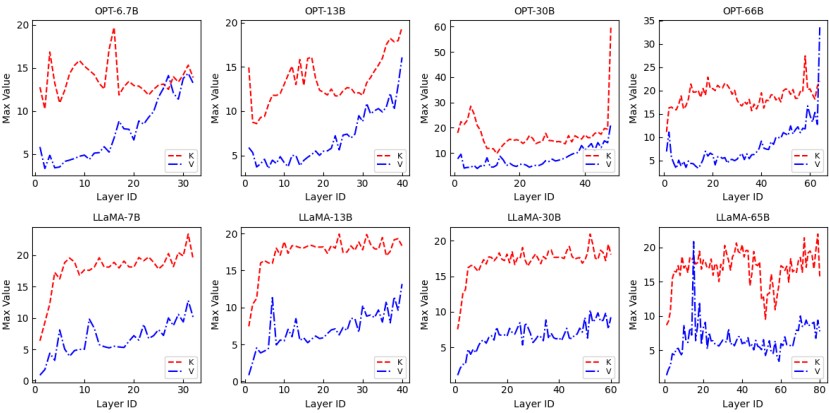

Figure 2: In this figure, we plot the maximum absolute value of the KV cache of each layer for OPT and LLaMA series models. The range of the key embeddings is larger than the range of the value embeddings.

The above analyses lead to different conclusions. Which factor is dominant needs to be verified experimentally. Our experimental results show that under the fine-grained symmetric dynamic quantization setting, the key embeddings deserve more bits. See section 4.3 for more experimental data.

### 3.5 QUANTIZING THE KV CACHE TO TERNARY DIGITS CAN REDUCE THE REQUIRED MULTIPLICATION

We give an tiny example in figure 3 to illustrate the calculation under different modes. To complete the computation show in figure 3a, 3 multiplications and 2 additions are required. When one operand of matrix multiplication is quantized to INT4/INT8, we have to dequantize the operand before matrix multiplication. To complete the computation show in Figure 3b, 3 additional multiplications are required to dequantize the quantized operand. But when one operand of matrix multiplication is quantized to ternary digits, dequantization is no longer needed and we can finish the matrix multiplication with addition and subtraction. One thing have to notice is that the quantization step $\Delta$ should be multiplied finally.

We visualize the computation flow of the attention block under different quantization mode in figure 4. Figure 4a shows the normal computational flow of one head of the multi-head attention block. When the shape of KV cache is $[B, NH, S, HS]$, for each new query, the required multiplication of Matmul 1 is $Mul_{M1} = B \times NH \times HS \times S$ and the required addition is $Add_{M1} = B \times NH \times (HS - 1) \times S$. The required multiplication of Matmul 2 is $Mul_{M2} = B \times NH \times HS \times S$ and the required adition is $Add_{M2} = B \times NH \times (S - 1) \times HS$.

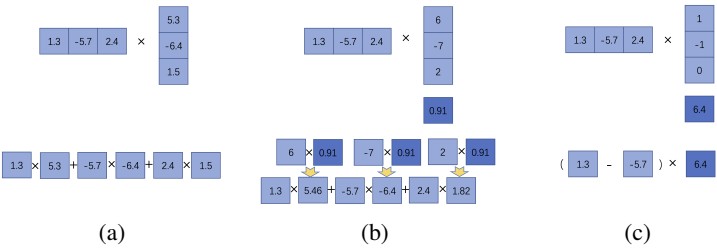

Figure 3: In this figure, we give a tiny example to illustrate the calculation under different modes. The numbers in the dark blocks are the $\Delta$ we use to do quantization.

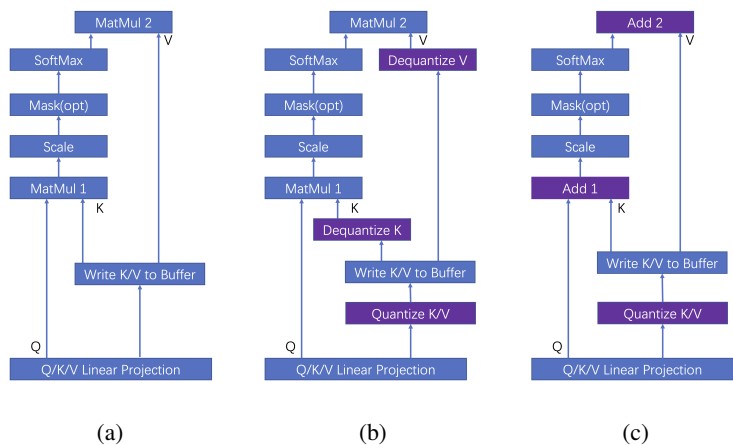

Figure 4: In this figure, we visualize the computation flow of the attention block under different setting. Figure 4a corresponds to the computation flow that no quantization operation is added to the KV cache. Figure 4b corresponds to the computation flow that quantizes the KV cache to INT4/INT8. Figure 4c corresponds to the computation flow that quantizes the KV cache to ternary digits.

As show in figure 4b, if we want to compress the KV cache to INT4/INT8, we have to add quantization node before writing the key-value embeddings to buffer and dequantize the KV cache to FP16 before using them. For text generation applications, the key-value embeddings are quantized once and the KV cache is dequantized multiple times. So the amount of calculation required for quantization is relatively small and can be ignored. Under the symmetric quantization setting, for a new query, the required calculation to dequantize the KV cache is $Mul_{DQ} = B \times NH \times S \times HS \times 2$.

As analyzed above, when compressing the KV cache to ultra-low bits, such as ternary digits, the dequantizing operation is no longer required and the multiplication operation in attention block can be replaced by addition, as show in figure 4c. In the inference process, for each new query vector, the $q \cdot k$ calculation is converted to adding the corresponding values of $q$ according to the value of $k$, since $k$ only contains $-1, 0, 1$. For a new query, the required calculation of Add 1 is $Add_{A1} = B \times NH \times (HS - 1) \times S$. There are still a few multiplication operations which are used to zoom the accumulation result using the quantization step $\Delta$. And the required calculation of Add 2 is $Add_{A2} = B \times NH \times (S - 1) \times HS$. Quantizing the KV cache to ternary digits can save $B \times NH \times S \times HS \times 2$ multiplication in dequantization and $B \times NH \times HS \times S \times 2$ multiplication in matrix multiplication. Please refer to Appendix A.2 for more detailed analysis.

## 3.6 KVTQ

Directly quantizing the KV cache to single channel of ternary digits result in unacceptable performance degradation. ABC-Net (Lin et al., 2017) use the multiple groups idea to alleviate the accuracy

drop when quantizing models to binary values. Based on the previous analysis of the KV cache and the multiple groups idea, we proposed the KVTQ method.

The KVTQ method has the following features: (1) fine-grained dynamic quantization, specifically, in our scenario we set the granularity to the head size of the multi-head attention. (2) symmetric quantization, which is friendly for quantizing the KV cache into compute efficient ternary digits. (3) ternary digits, which can reduce the required multiplication calculations. (4) multiple channels of ternary digits, the combination of multiple channels of ternary values of different quantization step can alleviate the accuracy degradation introduced by quantization. Combined with our sensitivity analysis of the KV cache, we use 4 channels of ternary digits for the key embeddings and 3 channels of ternary digits for the value embeddings.

The ternary digits with the largest quantization step can be obtained by equation 4, which is a special case of equation 3. It is clear that $\mathbf{X}^{q1}$ can only consist of $-1, 0, 1$.

$$\mathbf{X}^{q1} = \lceil \frac{\mathbf{X}}{\Delta_1} \rfloor, \Delta_1 = max(|\mathbf{X}|) \tag{4}$$

The ternary digits with the second largest quantization step can be obtained by equation 5. And the round-off error caused by equation 4 is compensated in equation 5.

$$\mathbf{X}^{q2} = \lceil \frac{\mathbf{X} - \mathbf{X}^{q1} \times \Delta_1}{\Delta_2} \rfloor, \Delta_2 = max(|\mathbf{X} - \mathbf{X}^{q1} \times \Delta_1|) \tag{5}$$

The following ternary digits with smaller quantization steps can be calculated in a similar way.

## 4 EMPIRICAL EVALUATION

In this section, we first introduce our experimental setup. Next we show that our KVTQ method can successfully compress the KV cache into ternary digits with negligible accuracy degradation. Then we experimentally demonstrate that the key embeddings are more sensitive to quantization. Finally we verify that using multi-channels of ternary digits to represent the KV cache does not significantly increase the addition operations required by the attention block.

### 4.1 EXPERIMENTAL SETUPS

Perplexity is widely used to evaluate the performance of the quantization methods for LLMs. We also use the perplexity to evaluate our KVTQ method. We refer to the Huggingface tutorial [3] to implement the calculation of perplexity and the only change is that we set $stride = max\_length$. Using the above mentioned implementation, the perplexity baseline calculated at FP16 precision is consistent with the results given in (Frantar et al., 2022; Yao et al., 2023).

In our experiments, we use two series of models of LLMs: OPT (Zhang et al., 2022) and LLaMA (Touvron et al., 2023). In order to compare and display our experimental conclusions more conveniently, we took the intersection based on the model sizes of the two series of models. We choose the most commonly used dataset, the test set of wikitext-2 (Merity et al., 2016) as the evaluation dataset.

### 4.2 QUANTIZING THE KV CACHE TO TERNARY DIGITS CAN ALSO PRESERVE ACCURACY

Compressing the KV cache have been mentioned in (Sheng et al., 2023; Liu et al., 2023), but they only gave limited experimental results. So we implement the per-tensor static quantization used in TensorRT-LLM and fine-grained dynamic quantization as contrasting methods. As we know, we also the first use fine-grained dynamic quantization to quantizing the KV cache to 4 bits/3 bits and systematically evaluate the performance.

In table 1, the TRT-LLM means using per-tensor static quantization to compress the KV cache to 4bit signed integers. K4 refers to quantizing of the K cache to 4 bits signed integers using per-head

---
[3]https://huggingface.co/docs/transformers/perplexity

symmetric dynamic quantization, and V4 refers to quantizing of the V cache to 4 bits signed integers using per-head symmetric dynamic quantization.

Table 1: Perplexity (lower is better) of the OPT and LLaMA series models on wikitext-2 dataset under different quantization settings.

| Model | Quantization Setting | 6.7B/7B | 13B | 30B | 65B/66B |
|-------|---------------------|---------|-----|-----|---------|
| OPT | FP16 | 10.8594 | 10.1172 | 9.5625 | 9.3395 |
| | TRT-LLM | 13.3047 | 12.9453 | 13.4844 | 14.9717 |
| | K4/V4 | **10.9219** | 10.2188 | **9.6016** | **9.3791** |
| | KVTQ | 10.9609 | **10.2188** | 9.6172 | 9.4430 |
| LLaMA | FP16 | 5.6719 | 5.0898 | 4.0977 | 3.5273 |
| | TRT-LLM | 74.9375 | 44.7500 | 16.4219 | 18.3906 |
| | K4/V4 | 5.8438 | 5.1992 | 4.1992 | 3.6055 |
| | KVTQ | **5.7930** | **5.1680** | **4.1758** | **3.5801** |

As show in table 1, using the per-tensor static quantization method to compress the KV cache to 4 bits integers will lead to unacceptable performance degradation. Our KVTQ method suffers negligible accuracy degradation compared to the FP16 baseline. On the newer LLaMA series models, which have better performance compared to the OPT series models, our KVTQ method is stably outperform the K4/V4 fine-grained dynamic quantization setting.

In addition to accuracy, performance also needs to be considered. Ternary digits are more suitable to be stored on ternary memory device (Zhang et al., 2019). The KVTQ method can be run on devices such as GPUs but such devices cannot take full advantage of the KVTQ method. The KVTQ method requires a customized ASIC to release its advantages. Ternary memory device plus addition unit can handle the storage and the usage of the KV cache. Such computing in memory technology has great potential when dealing with the KV cache, especially when the sequence length is very large. For LLM inference on devices such as GPUs, the K4/V4 method proposed in our paper is a better choice.

### 4.3 KEY CACHE REQUIRES MORE SPACE

According to the analysis in section 3.4, we designed a set of experiments to verify which one, K or V, is more sensitive to quantization. The experimental setting and results are summarized in table 2. Here K4/V4 have the same meaning as in section 4.2 and K3/K8/V3/V8 have the similar meaning. For more experimental data, please refer to Appendix A.1.

Table 2: Perplexity ↓ of the OPT and LLaMA series models on wikitext-2 dataset under conjugate quantization settings of the KV cache.

| Model | Quantization Setting | 6.7B/7B | 13B | 30B | 65B/66B |
|-------|---------------------|---------|-----|-----|---------|
| OPT | K8/V4 | 10.8594 | 10.1406 | 9.5625 | 9.3455 |
| | K4/V8 | 10.8984 | 10.2188 | 9.6016 | 9.3725 |
| | K4/V3 | 10.9844 | 10.2578 | 9.6406 | 9.4153 |
| | K3/V4 | 11.4922 | 11.4453 | 9.9453 | 10.2558 |
| LLaMA | K8/V4 | 5.6992 | 5.0977 | 4.1133 | 3.5352 |
| | K4/V8 | 5.8125 | 5.1875 | 4.1758 | 3.5977 |
| | K4/V3 | 5.9258 | 5.2695 | 4.2539 | 3.6680 |
| | K3/V4 | 7.3594 | 6.1797 | 5.2383 | 4.3594 |

From table 2, it is clear that the key embeddings are more sensitive to quantization, especially when compressed to ultra-low bits. When the total compression ratio is fixed, it would be better to allocate more space to the K cache. This conclusion guides us to choose a more appropriate configuration of the number of channels of ternary digits when compressing the KV cache.

## 4.4 KVTQ DOES NOT INTRODUCE TOO MANY EXTRA ADDITION OPERATIONS

In our KVTQ method, we use 4 channels of ternary digits to represent the K cache, and 3 channels of ternary digit to represent the V cache. According to analysis in section 3.5, the required addition operations is 4 times and 3 times compared to the original mode. But actually the sparseness of the ternary digits can offset the additional addition operations introduced by using multiple channels of ternary digits in KVTQ.

Table 3: Sparse ratio of the ternary feature maps of the OPT and LLaMA series models evaluated on wikitext-2 dataset. Channel 1 is the channel with largest quantization step and channel 2 is the channel with second largest quantization step and so forth.

| Model | Sparsity ratio / Channel | 6.7B/7B | 13B | 30B | 65B/66B |
|---|---|---|---|---|---|
| OPT | channel 1 of K | 0.24 | 0.27 | 0.13 | 0.43 |
| | channel 2 of K | 0.49 | 0.47 | 0.33 | 0.52 |
| | channel 3 of K | 0.50 | 0.49 | 0.43 | 0.52 |
| | channel 4 of K | 0.49 | 0.49 | 0.47 | 0.50 |
| | Sum of K | 1.72 | 1.72 | 1.36 | 1.97 |
| | channel 1 of V | 0.13 | 0.14 | 0.15 | 0.15 |
| | channel 2 of V | 0.39 | 0.40 | 0.42 | 0.42 |
| | channel 3 of V | 0.49 | 0.49 | 0.49 | 0.49 |
| | Sum of V | 1.01 | 1.03 | 1.06 | 1.07 |
| LLaMA | channel 1 of K | 0.03 | 0.03 | 0.03 | 0.07 |
| | channel 2 of K | 0.15 | 0.14 | 0.14 | 0.21 |
| | channel 3 of K | 0.35 | 0.34 | 0.33 | 0.38 |
| | channel 4 of K | 0.46 | 0.46 | 0.46 | 0.47 |
| | Sum of K | 0.99 | 0.97 | 0.96 | 1.13 |
| | channel 1 of V | 0.15 | 0.15 | 0.15 | 0.15 |
| | channel 2 of V | 0.41 | 0.42 | 0.42 | 0.42 |
| | channel 3 of V | 0.48 | 0.49 | 0.49 | 0.49 |
| | Sum of V | 1.04 | 1.06 | 1.06 | 1.06 |

In table 3, we counted the sparsity of different channels of the ternary digits used to express the KV cache. For both series of models, in calculations using the V cache, the KVTQ method reduces the number of multiplication operations while causing almost no increase in the amount of addition operations. For the K cache, the sparsity ratio of the two series of models are different. For OPT series models, the required addition operations is about 1.69 times of the original mode in calculations using the K cache. For LLaMA series models, it is only 1.01 times. The KVTQ method successfully replaces the original huge number of multiplication operations with a small number of additional addition operations.

## 5 CONCLUSION AND FUTURE WORK

We propose the KVTQ method, which can compressing the KV cache to hardware efficient ternary digits with negligible accuracy degradation. For the KVTQ method, the matrix multiplication operation in the attention block is replaced by addition. This innovation supports us to develop efficient heterogeneous devices for the storage and the usage of the KV cache. Besides, the numerical characteristics of the KV cache we discovered can serve as a reference for others studying the KV cache compression.

Future work includes combining the KVTQ method with pruning of the KV cache, such as SCISSORHANDS (Liu et al., 2023) and SparseGPT (Frantar & Alistarh, 2023). Furthermore, in the KVTQ method we use a fixed granularity to quantize the KV cache and a fixed number of ternary digits to represent the compressed KV cache. It is clear that the KV cache of different layers can be quantized with different granularities and represnetd with different number of ternary digits to achieve the best balance of accuracy and compression ratio. These two variables can be automatically selected through methods such as AutoML (He et al., 2021).

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

## A  APPENDIX

### A.1  ADDITIONAL DATA OF KV SENSITIVITY COMPARISON

In table 4, we give more data to show that the key embeddings are more sensitive to quantization than the value embeddings.

Table 4: Perplexity ↓ of the OPT and LLaMA series models on wikitext-2 dataset under partial quantization settings of the KV cache.

| Model | Quantization Setting | 6.7B/7B | 13B | 30B | 65B/66B |
|-------|---------------------|---------|-----|-----|---------|
| OPT | K4 | 10.8984 | 10.1953 | 9.6016 | 9.3728 |
| | V4 | 10.8594 | 10.1406 | 9.5625 | 9.3459 |
| LLaMA | K4 | 5.8125 | 5.1875 | 4.1758 | 3.5977 |
| | V4 | 5.6992 | 5.0977 | 4.1133 | 3.5352 |

In table 5, K4 refers to quantizing of the K cache to 4 bits signed integers using per-token symmetric dynamic quantization, V4 refers to quantizing of the V cache to 4 bits signed integers using per-token symmetric dynamic quantization, and K8/V8 means similar operation. The experimental data shows that when the KV cache is quantized with coarser granularity, the conclusion that the key embeddings are more sensitive to quantization still holds.

Table 5: Perplexity ↓ of the OPT and LLaMA series models on wikitext-2 dataset under per-token quantization settings of the KV cache.

| Model | Quantization Setting | 6.7B/7B | 13B | 30B | 65B/66B |
|-------|---------------------|---------|-----|-----|---------|
| OPT | K4 | 11.0938 | 10.2969 | 9.7500 | 9.6700 |
| | V4 | 10.9219 | 10.2422 | 9.6406 | 9.4095 |
| | K4/V8 | 11.0938 | 10.2812 | 9.7344 | 9.6716 |
| | K8/V4 | 10.9219 | 10.2422 | 9.6406 | 9.4097 |
| LLaMA | K4 | 5.9531 | 5.3047 | 4.2773 | 3.6719 |
| | V4 | 5.7930 | 5.1641 | 4.2461 | 3.6543 |
| | K4/V8 | 5.9531 | 5.3047 | 4.2461 | 3.6719 |
| | K8/V4 | 5.7930 | 5.1641 | 4.2461 | 3.6582 |

## A.2 THE NUMBER OF MULTIPLICATION OPERATIONS SAVED BY KVTQ

In section 3.5 we evaluate the amount of multiplication operations reduced by the KVTQ method, but due to limited space we did not give a complete example.

Here we use LlaMA-7B as an example. For LlaMA-7B the head number is 32 and the head size is 128. If we assume the input sequence length is 1024 and the maximum sequence length is 4096, for the generation process, the average sequence sequence will be $(1024 + 4096)/2$. As mentioned in section 3.5, the KVTQ method reduce $4 \times B \times NH \times S \times HS$ times of multiplication operations. If we assume the batch size is 16, in the generation process the reduced multiplication operations of one layer of the LlaMa-7B model are:

$$4 \times 16 \times 32 \times (1024 + 4096)/2 \times 128 \times (4096 - 1024) = 1.875T$$

For the LlaMA-7B model with 32 transformer blocks, the total reduced multiplication operations are:

$$1.875T \times 32 = 60T$$

In table 6, we give the amount of multiplication operations saved by the KVTQ method for OPT and LLaMA series models under the same setting mentioned above.

Table 6: The number of multiplication operations saved by KVTQ.

| Model | 6.7B/7B | 13B | 30B | 65B/66B |
|-------|---------|-----|-----|---------|
| OPT | 60T | 93.75T | 157.5T | 270T |
| LLaMA | 60T | 93.75T | 182.8125T | 300T |

