# OpenReview forum: "KVTQ: Compressing the KV Cache to Hardware Efficient Ternary Digits by Fine-Grained Dynamic Quantization"
_ICLR.cc/2024/Conference — Submitted to ICLR 2024_

### Official Review · Reviewer_nvar · 2023-10-28

**Soundness:** 2 fair
**Presentation:** 2 fair
**Contribution:** 3 good
**Rating:** 5
**Confidence:** 4

**Summary:**

This paper proposes a dynamic quantization technique to reduce the computational cost and required memory for storing K/V cache on GPU. The authors propose a quantization method that symmetrically quantizes the KV cache into ternary digits (-1, 0, 1), thereby obviating the need for a dequantization stage, which is required in conventional methods.  The experimental results demonstrate the efficacy of their quantization technique, showcasing lower perplexity compared to alternative methods and a reduction in computational workload.

**Strengths:**

The authors present pioneering research by demonstrating that KV caches can be quantized into ternary digits with minimal impact on perplexity. This innovative approach not only eliminates the need for a dequantization stage but also offers the benefits of reduced GPU memory usage and computational complexity.

**Weaknesses:**

- The authors present statistics regarding the maximum and minimum values of the KV cache in each layer, but do not delve into the distribution of data within each KV cache. While data distribution may be less relevant when quantizing data into n-bit integers, this paper opts for ternary digit quantization. Consider a key embedding to follow a normal distribution with the distance between the average and maximum value being 3 times the standard deviation (Max = $\mu + 3\sigma$). According to the authors' equation ($X_q = \lceil \frac{X}{\Delta} \rfloor, \Delta = \text{max}(|X|)$, approximately 86.6% of the numbers will be quantized to zero ($P(-1.5<\frac{x-\mu}{\sigma}<1.5)$). Such a substantial portion of key embeddings being quantized to zero might suggest sparsity in attention scores. Hence, it would be more persuasive if the authors compared their work against models employing sparse attention, as opposed to traditional attention mechanisms.

- The claim that quantizing to ternary digits reduces the number of multiplications and additions should be validated through runtime measurements, particularly when considering that LLM inference is primarily constrained by memory bandwidth and latency. It would be also advisable for the authors to measure runtime performance compared to other works, such as TRT-LLM. Additionally, as the authors propose a dynamic quantization approach, it is implied that the KV cache should still be stored in its original precision (e.g., FP32) and quantized before calculating attention scores. In contrast, TRT-LLM, one of the baseline models, employs static quantization, eliminating the need for repetitive quantization, and allows for lower precision storage in CPU/GPU memory. Consequently, the authors should address the potential overhead in terms of memory usage and quantization latency.

- There are also some minor suggestions:
  - Many sentences begin with conjunctions such as "And" and "So." To maintain formality, it is recommended to avoid starting sentences with conjunctions.

  - A minor typographical error is present in Section 1, line 2 ("konw" should be corrected to "known"). Moreover, the terms "Experiment setup/result" should be revised to "Experimental setup/result" for clarity and consistency.

**Questions:**

- Would the utilization of a sparse attention method yield superior results compared to KVTQ in terms of perplexity, runtime, and memory usage?
- Could you provide insights into the performance enhancements achieved with KVTQ when compared to other baseline methods, particularly in terms of runtime and memory usage?

---

> ### Author Response · Authors · 2023-11-16
>
> Thank you for your detailed comments!
>
> ---
>
> **KVTQ vs naive sparsity(item 1)**
>
> As show in table 3, we evaluate the sparse ratio of the KV cache of difference series of
> models under our KVTQ quantization setting.
> The KV cache of different models has different sparse ratio after quantization.
> The sparse ratio of the KV cache with the largest quantization step is high as your
> analyze.
> But if we only use one channel of ternary digit to express the KV cache, the accuracy is
> unacceptable.
> So we use 4 channels of ternary digits with different quantization step to express the K
> cache and 3 channels for the V cache.
> The sparse ratio of the KV cache with the smallest quantization step is about 50%.
>
>
> Scissorhands [1] compresses the KV cache by only saving the key-value embeddings with
> high attention scores, which is one kind of sparsity.
> They give the perplexity result as a curve in a figure.
> It can be seen from the figure that when the sparse ratio is greater than 50%, ppl
> increases significantly.
>
> **Performance(item 2)**
>
>
> Ternary digits are more suitable to be stored in ternary memory device.
> On existing LLM inference devices, such as GPUs, the KVTQ method can be run.
> But the KVTQ method requires a customized ASIC to release its advantages.
> For example, the KVTQ method can be combined with ternary memory device and computing in
> memory technology to avoid the cost of moving the KV cache.
>
>
> For LLM inference on devices such as GPUs, the K4/V4 method proposed in our paper is a
> better choice.
> As we know, we also the first use fine-grained dynamic quantization to quantizing the KV
> cache to 4 bits/3 bits and systematically evaluate the performance.
> We wanted to highlight the KVTQ method so we did not list the performance data of the
> K4/V4 method.
>
> We will revise the paper to emphasize that the advantages of KVTQ requires using ASIC to
> release.
>
>
> **Type and presentation(item 3)**
>
> Thank you, we will fix the typo error and refine our presentation follow your suggestions.
>
>
> ---
>
> If any questions remain, we are happy to engage in further discussion!
>
>
>
> **Reference**
>
> - [1] Zichang Liu, Aditya Desai, Fangshuo Liao, Weitao Wang, Victor Xie, Zhaozhuo Xu, Anastasios
> Kyrillidis, and Anshumali Shrivastava. Scissorhands: Exploiting the persistence of importance
> hypothesis for llm kv cache compression at test time. arXiv preprint arXiv:2305.17118, 2023

---

> ### Comment · Area_Chair_5trx · 2023-12-04
> **[Important] Response Required to Authors' Rebuttal**
>
> Dear Reviewer nvar,
>
> As we progress through the review process for ICLR 2024, I would like to remind you of the importance of the rebuttal phase. The authors have submitted their rebuttals, and it is now imperative for you to engage in this critical aspect of the review process.
>
> Please ensure that you read the authors' responses carefully and provide a thoughtful and constructive follow-up. Your feedback is not only essential for the decision-making process but also invaluable for the authors.
>
> Thank you,
>
> ICLR 2024 Area Chair

---

### Official Review · Reviewer_iUPn · 2023-10-31

**Soundness:** 3 good
**Presentation:** 3 good
**Contribution:** 3 good
**Rating:** 6
**Confidence:** 2

**Summary:**

This paper proposes a novel fine-grained dynamic quantization method, named KVTQ, for compressing the Key-Value cache of LLMs into hardware-efficient ternary digits. The authors highlight that, unlike traditional weights-only quantization, which requires dequantization for each use, their method eliminates the need for dequantization and allows matrix multiplication to be efficiently conducted using simple addition and subtraction. They claim to be the first to demonstrate that compressing the KV cache to ternary digits can be achieved with negligible impact on perplexity.

**Strengths:**

The motivation behind employing KVTQ (Key-Value Token Quantization) is clearly articulated and easy to grasp. The paper is well-written, with a logical flow that makes it easy to follow the core concept. However, I must admit that my expertise may not fully equip me to assess the technical novelty in this particular field.

From what I understand, a key aspect of KVTQ is its ability to avoid additional dequantization steps. Instead, it can directly replace dequantization and subsequent matrix multiplication with a summation operation. This approach seems to have practical implications, particularly in reducing computational complexity.

The empirical results demonstrate a significant reduction in perplexity across various sizes of OPT/LLama models, which is noteworthy given that many expensive operations are bypassed. This aspect of KVTQ seems to be a valuable contribution, potentially leading to more efficient processing in relevant applications. However, a deeper technical analysis might be necessary to fully appreciate the novelty and implications of this approach.

**Weaknesses:**

I confess that my understanding of quantization isn't particularly deep, which somewhat hinders my ability to fully grasp the implications of the results shown in Tables 3 and 4.
However, I think the paper falls short in providing comprehensive information on the practical aspects of implementing KVTQ. Details like the physical size of the Key-Value (KV) cache when KVTQ is in use, as well as the memory overhead and latency during forward passes, are missing. These details are crucial for understanding not just the theoretical benefits of KVTQ, but also its real-world applicability and efficiency.

**Questions:**

see weaknesses.

---

> ### Author Response · Authors · 2023-11-16
>
> Thank you for your comments!
>
> ---
>
> Table 3 shows the sparse ratio of the KV cache which have been quantized using the KVTQ
> method.
> The sparsity feature of the quantized KV cache means that we use multiple channels of
> ternary digits to represent the KV cache does not result in too many additional additive
> calculation while eliminating multiplication calculations.
>
> Table 4 is a supplementary experiment to illustrate that K cache is more sensitive to
> quantization than V cache.
> This conclusion supports that we give K cache more bits when quantizing KV cache.
>
> Ternary digits are more suitable to be stored in ternary memory device.
> On existing LLM inference devices, such as GPUs, the KVTQ method can be run.
> But the KVTQ method requires a customized ASIC to release its advantages.
> For example, the KVTQ method can be combined with ternary memory device and computing in
> memory technology to avoid the cost of moving the KV cache.
>
> For LLM inference on devices such as GPUs, the K4/V4 method proposed in our paper is a
> better choice.
> As we know, we also the first use fine-grained dynamic quantization to quantizing the KV
> cache to 4 bits/3 bits and systematically evaluate the performance.
> We wanted to highlight the KVTQ method so we did not list the performance data of the
> K4/V4 method.
>
> We will revise the paper to emphasize that the advantages of KVTQ requires using ASIC to
> release.
>
>
> ---
>
> If any questions remain, we are happy to engage in further discussion!

---

> > ### Comment · Reviewer_iUPn · 2023-11-22
> > **responding to authors**
> >
> > Thank you for your response. I will maintain my score.

---

> ### Comment · Area_Chair_5trx · 2023-12-04
> **[Important] Detailed feedback required**
>
> Dear Reviewer iUPn,
>
> I noticed that while you have opted to maintain your initial score, there were no detailed reasons accompanying this decision. It would be highly beneficial for both the authors and the integrity of the review process if you could provide more explicit reasoning behind your decision to maintain the current score. This detailed feedback is essential for the authors to understand the strengths and weaknesses of their work as perceived by the reviewers.
>
> Thank you,
>
> ICLR 2024 Area Chair

---

### Official Review · Reviewer_cw8R · 2023-10-31

**Soundness:** 2 fair
**Presentation:** 3 good
**Contribution:** 2 fair
**Rating:** 3
**Confidence:** 5

**Summary:**

This paper examines KV cache compression algorithms, specifically focusing on ternary representation. The ternary format streamlines computations by eliminating the need for a dequantization step and primarily utilizing addition and subtraction operations. The authors explore distinct quantization sensitivities for 'K' and 'V', each quantized with varying bit numbers. The study evaluates the LLaMA and OPT models and delves into the sparsity arising from the ternary representation.

**Strengths:**

- The authors provide detailed results concerning the quantization of K and V bits. By monitoring the range of K and V values across various LLaMA and OPT models, the authors validate the rationale for allocating distinct quantization bits to K and V.

- The paper highlights the unique computational advantages of ternary representation. Contrary to recent weight-only quantizations, ternary representation simplifies computations to mainly additions and subtractions.

- The study showcases sparsity across different channels, illustrating the potential computational savings from '0' weights in ternary quantization.

- The presented quantization techniques and computational approaches are clear and uncomplicated.

**Weaknesses:**

- While ternary computations can simplify attention-related calculations, the authors haven't quantified the reduction in latency or the number of FLOPs saved by their method.

- Given that ternary representation uses 2 bits to represent -1, 0, or +1, its memory footprint might surpass binary-based quantization. A comparison of memory usage between previous quantization methods and the proposed approach is essential.

- The correlation between sparsity and computational reduction doesn't directly equate to reduced latency or improved throughput. Instead of merely highlighting channel sparsity, tangible hardware benefits should be assessed.

- What is the net effect on inference? The paper would benefit from a thorough estimation or actual measurement results.

**Questions:**

Please refer to the list of weakness above.

---

> ### Author Response · Authors · 2023-11-16
>
> Thank you for your detailed comments!
>
> ---
>
> **Flops(item 1)**
>
> In section 3.5 we describe the amount of multiplication operations in the attention
> block, but due to limited space we did not give a complete example.
> We will refine this part and add more results in appendix.
>
> Here we use LlaMa-7B as an example:
>
> For LlaMa-7B the head number is 32 and head size is 128.
> If we assume the input sequence is 1024 and the maximum sequence length is 4096 and the
> average sequence will be $(1024 + 4096) / 2$.
> As mentioned in section3.5, the KVTQ method reduce
> $4 \times B \times NH \times S \times HS$ times of multiplication operations.
>
> If we assume the batch size is 16, in the generation process the reduced multiplication
> operations of one layer of the LlaMa-7B  model are:
> $$ 4 * 16 * 32 * (1024 + 4096) / 2 * 128 * (4096 - 1024) = 1.875T $$
> For the LlaMa-7B model with 32 transformer blocks, the total reduced multiplication
> operations are:
> $$ 1.875T * 32 = 60T $$
>
>
>
> **Memory usage(item 2)**
>
> In KVTQ, we use 4 groups of ternary digits to express the K cache and 3 groups of ternary
> digits to express the V cache.
> As you said, if we want to storage the ternary K cache in binary memory device directly,
> we have to use 8 bits and for V cache, we have to use 6 bits.
> The memory usage is less than the 8-bit static quantization used in TRT-LLM but the
> memory usage is higher than the K4/V4 fine-grained dynamic quantization we proposed in
> our paper as comparison method.
>
> Ternary digits are more suitable to be stored in ternary memory device.
> And saving memory is not the only target considered by the KVTQ method.
> The K4/V4 method we proposed in our paper saves more memory compared to the KVTQ method,
> but the K4/V4 method does not have the computing advantages which we mentioned in section
> 3.5.
>
>
> **Sparsity and performance(item 3)**
>
> As you said, sparsity and computational reduction doesn't directly equate to reduced
> latency or improved throughput.
> On existing LLM inference devices, such as GPUs, the KVTQ method can be run.
> But the KVTQ method requires a customized ASIC to release its advantages.
> For hardware such as GPU, the K4/V4 method we proposed in our paper is more suitable.
>
>
> **Over all performance(item 4)**
>
> Ternary digits are more suitable to be stored in ternary memory device.
> On existing LLM inference devices, such as GPUs, the KVTQ method can be run.
> But the KVTQ method requires a customized ASIC to release its advantages.
> For example, the KVTQ method can be combined with ternary memory device and computing in
> memory technology to avoid the cost of moving the KV cache.
>
> We will revise the paper to emphasize that the advantages of KVTQ requires using ASIC to
> release.
>
> ---
>
> If any questions remain, we are happy to engage in further discussion!

---

> > ### Comment · Reviewer_cw8R · 2023-11-21
> > **Response from Reviewer cw8R**
> >
> > Thank you for your detailed responses. However, I still have some critical concerns:
> >
> > 1. In highly parallelizable computing systems like GPUs, the direct relationship between reduced computational workload and improved latency or throughput is not always clear, especially when certain computations become less parallelizable. Therefore, the stated 60T reduction in multiplication operations doesn't clearly translate to more efficient inference.
> > 2. The concept of a ternary memory device is intriguing, but I'm uncertain about its practicality. As far as I'm aware, there aren't any commercial memory systems available that can distinctly recognize three values (such as -1, 0, +1) in one bit-cell memory structure. Could you clarify what you mean by this?
> > 3. The necessity for a customized ASIC warrants a more detailed explanation. What specific features of such ASIC are essential for your approach? If the focus is heavily on sophisticated hardware, it might be more appropriate to consider publishing this research in a hardware-oriented community.
> >
> > Given these unresolved issues, I have decided to maintain my original score.

---

> > > ### Author Response · Authors · 2023-11-22
> > >
> > > Thank you for your responses.
> > >
> > > ---
> > >
> > > **Flops**
> > >
> > > You mentioned the flops saved by the KVTQ method, so we made additional explanations on
> > > this point.
> > > For hardware such as GPU, the K4/V4 method we proposed in our paper is more suitable.
> > > Implementing the KVTQ method on GPU, considering parallelism and efficiency, we can not
> > > use the sparsity features of the quantized KV cache.
> > >
> > >
> > > **Ternary memory devices**
> > >
> > > Ternary memory device is widely studied [1, 2].
> > > What we want to express is that the ternary memory device has its value in caching the
> > > key-value embeddings of LLMs.
> > > We are not sure whether ternary memory devices will be commercially available in the
> > > future.
> > > But we will not limit our view just because the ternary memory device is not commercially
> > > available.
> > > If you focus on existing commercially available devices, such as GPUs, as we mentioned in
> > > our paper the K4/V4 method is a more suitable choice.
> > >
> > >
> > >
> > > **Algorithm rather than hardware**
> > >
> > > In this paper, we focus on the algorithm which can be used to quantize the KV cache to
> > > ultra-low bit digits rather than how to design an ASIC to support our algorithm.
> > > When designing hardware-friendly algorithms, we need to understand what operations are
> > > efficient for hardware but we don't think about how to design the hardware.
> > > Implementing a multiplier in hardware requires more NAND gates than implementing an adder.
> > > Completing a multiplication operation in hardware requires more clock cycles than
> > > completing an addition operation.
> > > Different devices require different clock cycles to complete a multiplication and
> > > addition operation, but generally speaking, an addition operation requires 1 clock cycle,
> > > while a multiplication operation usually requires 3-5 clock cycles or even more.
> > > Therefore, in the KVTQ method, in addition to compression, we focus on eliminating
> > > multiplication operations but we will not consider how to make an adder.
> > >
> > >
> > > ---
> > >
> > > If any questions remain, we are happy to engage in further discussion!
> > >
> > >
> > >
> > > **Reference**
> > >
> > > - [1] Hongyan Zhang, Xiaofeng Zhao, Ju Bai, Yanjun Hou, Shuhong Wang, Cheng Wang, and Dongge
> > > Ma. Ternary memory devices based on bipolar copolymers with naphthalene benzimidazole ac-
> > > ceptors and fluorene/carbazole donors. Macromolecules, 52(23):9364–9375, 2019.
> > > - [2] Hongtao Zhong, Shengjie Cao, Huazhong Yang, and Xueqing Li. Dynamic ternary content-
> > > addressable memory is indeed promising: Design and benchmarking using nanoelectromechani-
> > > cal relays. 2021.

---

> ### Comment · Area_Chair_5trx · 2023-12-04
> **[Important] Response Required to Authors' Rebuttal**
>
> Dear Reviewer cw8R,
>
> As we progress through the review process for ICLR 2024, I would like to remind you of the importance of the rebuttal phase. The authors have submitted their rebuttals, and it is now imperative for you to engage in this critical aspect of the review process.
>
> Please ensure that you read the authors' responses carefully and provide a thoughtful and constructive follow-up. Your feedback is not only essential for the decision-making process but also invaluable for the authors.
>
> Thank you,
>
> ICLR 2024 Area Chair

---

### Official Review · Reviewer_ZD7v · 2023-11-01

**Soundness:** 2 fair
**Presentation:** 2 fair
**Contribution:** 2 fair
**Rating:** 5
**Confidence:** 4

**Summary:**

The paper introduces a method to compress the KV-cache into ternary, effectively reducing both storage and computational costs. Specifically, regarding computational costs, the ternary KV-cache eliminates the need for reweighting, converting multiplications into addition and subtraction operations. The work uses experimental statistics to guide the ternarization process.

**Strengths:**

1.	Ternarization indeed results in a reduction in storage and computational costs.

2.	Experimental findings play an instructive role in quantization. The paper discovers that it's preferable to allocate more bits to K due to its larger numerical rang.

**Weaknesses:**

1.	The paper's description of the actual quantization method might lead to misleading. The term "ternary" suggests that K and V are genuinely ternary with values {-1, 0, +1}. However, the paper assigns "multiple channels" to each value with varying steps. This essentially equates to a higher quantization bit count. As the paper states, “we use 4 channels of ternary digits for the key embeddings and 3 channels of ternary digits for the value embeddings.” This means K is 4-bit quantized, and V is 3-bit quantized? This crucial point lacks adequate discussion and might mislead readers.
2.	The quantization method utilized is dynamic, meaning the quantization step must be dynamically determined. This approach may not be hardware-friendly. To determine a single max and min value requires scanning the entire tensor. For larger tensors, this method could introduce significant latency.
3.	The ternary compensation algorithm employed originates from ABC-Net and is not original.
4.	The paper lacks experimental details. While there were experiments on PPL, the motivation of “reducing storage and computational costs” is not reflected in the experiments. It remains unproven whether ternary quantization is GPU-friendly. Likewise, there's no evidence provided to demonstrate if dynamic quantization will introduce substantial latency.

**Questions:**

1.	Can you provide clarity on the choice of using "ternary" in your terminology when the actual quantization might suggest higher bit counts?
2.	How do you address the potential hardware inefficiencies of the dynamic quantization method, especially with larger tensors?
3.	Since the ternary compensation algorithm is taken from ABC-Net, how is your quantization method different from it?
4.	You should provide more experiments that can showcase the effectiveness of your method in terms of computational and storage savings. For example, how much decoding latency can be reduced?

---

> ### Author Response · Authors · 2023-11-16
>
> Thank you for your detailed comments!
>
> ---
>
> **Why we use "ternary"(Q1)**
>
> The ternary digits have two nice properties.
> First for $A \times B $, when either A or B contains only ternary digits, the matrix
> multiplication can be completed by addition and subtraction operations.
> 4 bit/3 bit integers do not have this feature.
> We give a tiny example in figure 3.
>
> Second the ternary representation will result in sparsity.
> We can take advantage of this feature to reduce the amount of calculations.
> 4 bit/3 bit integers do not have this feature.
> Figure 3c is an example.
>
> For these two reasons, we use “ternary” in our paper.
>
>
> **Overhead of dynamic quantization(Q2)**
>
>
> For text generation applications, the KV cache is quantized once and used multiple times.
> During the generation process, for each new query, we only have to quantize the key-value
> embeddings corresponding the current query since the key-value embeddings of the past
> querys have been cached.
> For each new query, the size of key-value embedding is $1 * hiddensize$.
>
> Compared to static quantization of the KV cache, the only difference is that we have to
> compute the quantization step during the inference process for dynamic quantization.
> The amount of calculation introduced by dynamic quantization is relatively small and can
> be ignored.
>
>
> **Relationship with ABC-Net(Q3)**
>
> Ternary quantization have been used in computer vision tasks formerly and we are the
> first which use this technique in quantizing the KV cache of LLMs.
> Previously, binary/ternary quantization was used within the scope of quantization-aware
> training(QAT).
> Our KVTQ applies ternary quantization under the scope of post-training quantization(PTQ).
>
> We mentioned ABC-Net for it gives us the idea to use the **multiple groups** idea rather
> than it use ternary quantization.
> ABC-Net try to quantize AI models of computer vision tasks to binary value.
> ABC-Net focuses on binary quantization and the method is under the QAT scope.
> We will refine the description of the relation between our work and ABC-Net.
>
>
> **Memory and performance(Q4)**
>
> Ternary digits are more suitable to be stored in ternary memory device.
> On existing LLM inference devices, such as GPUs, the KVTQ method can be run.
> But the KVTQ method requires a customized ASIC to release its advantages.
> For example, the KVTQ method can be combined with ternary memory device and computing in
> memory technology to avoid the cost of moving the KV cache.
>
>
> For LLM inference on devices such as GPUs, the K4/V4 method proposed in our paper is a
> better choice.
> We will revise the paper to emphasize that the advantages of KVTQ requires using ASIC to
> release.
>
> ---
>
> If any questions remain, we are happy to engage in further discussion!

---

> ### Comment · Area_Chair_5trx · 2023-12-04
> **[Important] Response Required to Authors' Rebuttal**
>
> Dear Reviewer ZD7v,
>
> As we progress through the review process for ICLR 2024, I would like to remind you of the importance of the rebuttal phase. The authors have submitted their rebuttals, and it is now imperative for you to engage in this critical aspect of the review process.
>
> Please ensure that you read the authors' responses carefully and provide a thoughtful and constructive follow-up. Your feedback is not only essential for the decision-making process but also invaluable for the authors.
>
> Thank you,
>
> ICLR 2024 Area Chair

---

### Official Review · Reviewer_gbCq · 2023-11-07

**Soundness:** 2 fair
**Presentation:** 2 fair
**Contribution:** 2 fair
**Rating:** 3
**Confidence:** 2

**Summary:**

This paper proposed to use ternary quantization, KVTQ, on the KV cache in large language model inference to compress the memory space and improve the attention computation efficiency. KVTQ uses a group of ternary digits of different quantization steps to express the KV cache to help alleviate the accuracy degradation of multiple channels. The experiments show that the proposed KVTQ can outperform 4-bit KV cache quantization.

**Strengths:**

+ This paper systematically studies the KV cache quantization settings, including dynamic/static quantization, symmetric/asymmetric quantization, and quantization precision difference for K and V cache.
+ The evaluation results of the proposed KVTQ are promising, especially on the newer large language models LLaMa.

**Weaknesses:**

- The novelty of the proposed ternary quantization is limited since it was first proposed by ABC-Net.
- This paper lacks measured memory usage and memory footprint of the KV cache for the proposed KVTQ method.
- This paper also lacks measured latency/throughput using the proposed KVTQ method. The actual improvement of replacing the multiplication in attention with addition using ternary digits is unclear.

**Questions:**

Please provide the experiment results mentioned in the weaknesses part.

---

> ### Author Response · Authors · 2023-11-16
>
> Thank you for your comments!
>
> ---
>
> **Relationship with ABC-Net(item 1)**
>
> Ternary quantization have been used in computer vision tasks formerly and we are the
> first which use this technique in quantizing the KV cache of LLMs.
> Previously, binary/ternary quantization was used within the scope of quantization-aware
> training(QAT).
> Our KVTQ applies ternary quantization under the scope of post-training quantization(PTQ).
>
> We mentioned ABC-Net for it gives us the idea to use the **multiple groups** idea rather
> than it use ternary quantization.
> ABC-Net try to quantize AI models of computer vision tasks to binary value.
> ABC-Net focuses on binary quantization and the method is under the QAT scope.
> We will refine the description of the relation between our work and ABC-Net.
>
> **Memory usage(item 2)**
>
> In KVTQ, we use 4 groups of ternary digits to express the K cache and 3 groups of ternary
> digits to express the V cache.
> If we want to store the ternary K cache in binary memory device directly, we have to use
> 8 bits and for V cache, we have to use 6 bits.
> Ternary digits are more suitable to be stored in ternary memory device.
>
> Saving memory is not the only target considered by the KVTQ method.
> The K4/V4 method we proposed in our paper saves more memory compared to the KVTQ method,
> but the K4/V4 method does not have the computing advantages which we mentioned in section
> 3.5.
>
>
> **Performance(item 3)**
>
> On existing LLM inference devices, such as GPUs, the KVTQ method can be run.
> But the KVTQ method requires a customized ASIC to release its advantages.
> During the generation process, for each new query we need to load all the cached KV
> embeddings into the computing unit.
> For example, the KVTQ method can be combined with ternary memory device and computing in
> memory technology to avoid the cost of moving the KV cache.
>
> For LLM inference on devices such as GPUs, the K4/V4 method proposed in our paper is a
> better choice.
> As we know, we also the first use fine-grained dynamic quantization to quantizing the KV
> cache to 4 bits/3 bits and systematically evaluate the performance.
>
> We will revise the paper to emphasize that the advantages of KVTQ requires using ASIC to
> release.
>
> ---
>
> If any questions remain, we are happy to engage in further discussion!

---

> ### Comment · Area_Chair_5trx · 2023-12-04
> **[Important] Response Required to Authors' Rebuttal**
>
> Dear Reviewer gbCq,
>
> As we progress through the review process for ICLR 2024, I would like to remind you of the importance of the rebuttal phase. The authors have submitted their rebuttals, and it is now imperative for you to engage in this critical aspect of the review process.
>
> Please ensure that you read the authors' responses carefully and provide a thoughtful and constructive follow-up. Your feedback is not only essential for the decision-making process but also invaluable for the authors.
>
> Thank you,
>
> ICLR 2024 Area Chair

---

### Author Response · Authors · 2023-11-20
**Paper Revision  & Discussion Reminder**

Dear reviewers,

We would like to thank you again for your useful comments and would like to note that we just posted a paper revision integrating your feedback. We highlight the modified parts in blue.

We would be very happy to engage in further discussion about this paper revision.

With best regards, the authors.

---

### Meta-Review · Area_Chair_5trx · 2023-12-06

**Metareview:**

The paper presents an approach to quantizing KV caches into ternary digits for Large Language Models (LLMs), which has the potential to reduce GPU memory usage and computational complexity. The average rating given by the reviewers is 4.4, which suggests significant concerns about the paper's content and methodology. While the idea of ternary digit quantization and its potential benefits are intriguing, the reviewers have identified several critical issues that undermine the paper's effectiveness and applicability.

Given the concerns raised by the reviewers, I am inclined to recommend rejection of the paper.

**Justification For Why Not Higher Score:**

The authors have offered explanations concerning several aspects of their paper, including the connection between their KVTQ method and ABC-Net, the memory demands and operational requirements of the KVTQ method, their concentration on algorithmic progressions rather than hardware specifics, and their approach to managing sparsity in the method. Despite these clarifications being acknowledged, significant concerns persist and could not adequately address the issues highlighted by the reviewers.

**Justification For Why Not Lower Score:**

N/A

---

### Decision · Program_Chairs · 2024-01-16

Reject